# Tuft and Cancer Stem Cell Marker DCLK1: A New Target to Enhance Anti-Tumor Immunity in the Tumor Microenvironment

**DOI:** 10.3390/cancers12123801

**Published:** 2020-12-17

**Authors:** Zhiyun Cao, Nathaniel Weygant, Parthasarathy Chandrakesan, Courtney W. Houchen, Jun Peng, Dongfeng Qu

**Affiliations:** 1Academy of Integrative Medicine, Fujian University of Traditional Chinese Medicine, Fuzhou 350122, China; caozhiyun0824@163.com (Z.C.); nweygant@gmail.com (N.W.); 2Fujian Key Laboratory of Integrative Medicine in Geriatrics, Fujian University of Traditional Chinese Medicine, Fuzhou 350122, China; 3Key Laboratory of Integrative Medicine of Fujian Province University, Fujian University of Traditional Chinese Medicine, Fuzhou 350122, China; 4Department of Medicine, University of Oklahoma Health Sciences Center, Oklahoma City, OK 73104, USA; parthasarathy-chandrakesan@ouhsc.edu (P.C.); Courtney-Houchen@ouhsc.edu (C.W.H.); 5Peggy and Charles Stephenson Cancer Center, Oklahoma City, OK 73104, USA; 6Department of Veterans Affairs Medical Center, Oklahoma City, OK 73104, USA

**Keywords:** DCLK1, tuft cells, cancer stem cells, microenvironment, immunotherapies

## Abstract

**Simple Summary:**

Doublecortin-like kinase 1 (DCLK1) is a tumor stem cell marker in colon, pancreatic, and potentially other cancers that has received wide attention recently. Aside from its role as a tuft cell marker in normal tissue and as a tumor stem cell marker in cancer, previous studies have demonstrated that silencing DCLK1 functionally reduces stemness, epithelial mesenchymal transition (EMT), and tumorigenesis in cancers. More recently, DCLK1′s role in regulating the inflammatory, pre-cancer, and tumor microenvironment including its ability to modulate immune cell mechanisms has started to come into focus. Importantly, clinically viable therapeutic means of targeting DCLK1 have finally become available in the form of kinase inhibitors, monoclonal antibodies, and chimeric antigen receptor T cells (CAR-T). Herein, we comprehensively review the mechanistic role of DCLK1 in the tumor microenvironment, assess the potential for targeting DCLK1 in colon, pancreatic and renal cancer.

**Abstract:**

Microtubule-associated doublecortin-like kinase 1 (DCLK1) is an accepted marker of tuft cells (TCs) and several kinds of cancer stem cells (CSCs), and emerging evidence suggests that DCLK1-positive TCs participate in the initiation and formation of inflammation-associated cancer. DCLK1-expressing CSCs regulate multiple biological processes in cancer, promote resistance to therapy, and are associated with metastasis. In solid tumor cancers, tumor epithelia, immune cells, cancer-associated fibroblasts, endothelial cells and blood vessels, extracellular matrix, and hypoxia all support a CSC phenotype characterized by drug resistance, recurrence, and metastasis. Recently, studies have shown that DCLK1-positive CSCs are associated with epithelial-mesenchymal transition, angiogenesis, and immune checkpoint. Emerging data concerning targeting DCLK1 with small molecular inhibitors, monoclonal antibodies, and chimeric antigen receptor T-cells shows promising effects on inhibiting tumor growth and regulating the tumor immune microenvironment. Overall, DCLK1 is reaching maturity as an anti-cancer target and therapies directed against it may have potential against CSCs directly, in remodeling the tumor microenvironment, and as immunotherapies.

## 1. Introduction

Microtubule-associated doublecortin-like kinase 1 (DCLK1) was originally thought to be a brain-specific protein before 2006 [1] when Giannakis et al. first reported DCLK1 as a potential marker of stem-like cells of the small intestine [2]. However, further research has identified these cells as differentiated tuft cells (TCs) possessing a variety of unique molecular and functional characteristics [3]. DCLK1+ tuft cells of the gastrointestinal tract are characterized by microvilli and may be long-lived and display self-renewal or progenitor functionality under some conditions [4,5,6]. Importantly, they regulate the immune microenvironment through IL-25/IL-17RB signaling in order to affect epithelial repair after injury, and may initiate inflammation-associated tumorigenesis after mutation [7,8,9,10,11,12]. In 2008, the Houchen group proposed that DCLK1 is a specific marker protein for intestinal adenoma stem cells [13], which brought attention to DCLK1 in cancer research and was the first of a series of research reports providing evidence that it might be an effective target for oncology drug development. To date, DCLK1 has been demonstrated to be a relatively selective marker of several kinds of cancer stem-like cells or cancer stem cells (CSCs) including in colon, breast, pancreas, kidney, and esophageal cancers [14,15,16,17]. After twenty years of research, DCLK1 is accepted as a specific marker of TCs and several kinds of CSCs, and is well known for its ability to regulate tumor growth, invasion, metastasis, epithelial-mesenchymal transition (EMT), pluripotency, angiogenesis, and pro-survival signaling [18,19,20,21].

CSCs are an important subpopulation of cells in the immunosuppressive tumor microenvironment (TME), which in turn provides a niche to support stem cell characteristics including self-renewal, differentiation, and immunosuppressive cell recruitment. Tumors create an immunosuppressive microenvironment by secreting a variety of chemokines and cytokines which may recruit tumor associated macrophages (TAM), tumor associated neutrophils (TAN), myeloid derived suppressor cells (MDSC), and other regulatory immune cells. TAM and TAN differentiate from polarized macrophages and neutrophils respectively, and remodel the TME to support tumor growth and angiogenesis [22]. TAM have been shown to promote the degradation of extracellular matrix and secrete exosomes containing mRNA and miRNA which ultimately promote tumor invasion and metastasis. Both TAM and CD4+ T-cells secrete tumor necrosis factor alpha (TNF-α) and up-regulate NF-κB signal pathway to induce the expression of EMT transcription factors Snail and Twist [23]. Moreover, they enhance transforming growth factor-β (TGF-β) signaling to promote the self-renewal of CSCs [24]. Presently, CSCs are considered a key driver of chemotherapy resistance, recurrence, and metastasis. Recent work shows that DCLK1 promotes CSC self-renewal and drug-resistance and can be targeted to inhibit tumorigenesis in kidney cancer [25]. Furthermore, several recent studies show that DCLK1 affects tumor growth and metastasis via regulating TAM and immune checkpoint. Finally, monoclonal antibodies and chimeric antigen receptor T-Cells (CAR-T) based on DCLK1 have demonstrated potential as novel cancer immunotherapies [26,27,28]. Herein we review key advances in the understanding of DCLK1 and DCLK1+ TCs function in the context of the tumor and immune microenvironment and discuss future directions for DCLK1-based research and development.

## 2. Function of DCLK1-Expressing Gastrointestinal Tuft Cells

TCs are present above the +4 position of the intestinal crypt and in the villus where they function as a chemosensory and secretory cell type. Additionally, TCs are found in the respiratory tract, salivary gland, gallbladder, pancreatic duct, auditory tube, urethra, and thymus [29,30,31,32,33]. The majority within the intestinal epithelium express DCLK1, and accumulating evidence suggests that DCLK1+ tuft cells take part in a diffuse chemosensory system where they serve a sentinel function to detect chemical signals in the microenvironment and orchestrate the repair of local epithelial tissue [12]. For instance, TCs located in the lung, colon and stomach epithelium can sense alterations to pH, nutrients, or the microbiota using taste receptors including GTP-binding protein α-gustducin and transient receptor potential cation channel subfamily M member 5 (TRPM5), or regulate capillary resistance to hypoxia by inducing an epithelial response via secretion of IL-25, leading to the activation of innate lymphoid type 2 cells (ILC2) and IL-13 secretion [34,35,36,37]. Using a transgenic intestinal epithelium specific DCLK1 knockout mouse model (Villin^Cre^;Dclk1^fl/fl^), May et al. reported that DCLK1 deletion in tuft cells resulted in altered gene expression in pathways for epithelial growth, stemness, barrier function, and taste reception signaling further suggesting its importance in TCs [9]. Furthermore, several studies have reported that DCLK1-expressing TCs secrete various kinds of regulatory molecules such as leukotrienes, prostaglandins, nitric oxide and IL-25, which lead to ILC2 and LGR5+ stem cell-mediated tuft and goblet cell differentiation in chronic inflammation and injury [35,38]. This is a key emerging area of interest that will provide new knowledge about the inflammatory, pre-cancer, and tumor microenvironments as well as immune–tumor interactions as they relate to tumorigenesis and progression.

There is strong evidence that DCLK1 expression in TCs play an important functional role in epithelial repair processes of the gut. Intestinal epithelium-specific knockout of DCLK1 (Vil^Cre^;Dclk1^flox/flox^) leads to increased severity of injury and death in mouse whole body irradiation and dextran sulfate sodium (DSS)-induced colitis models [6,9,39]. A recent study expounded on this idea more directly. Yi et al. reported the deletion of DCLK1 in the mucin-type O-glycan deficient model of ulcerative colitis (UC) resulted in greater severity of disease characterized by enhanced mucosal thickening and increased inflammatory cell infiltration. They found that in the absence of DCLK1, epithelial proliferative responses to chronic inflammation were impaired. However, the deletion of DCLK1 did not affect the numbers of intact TCs. These results indicate that DCLK1 expression is a regulator of TC activation status, despite not being involved in TC expansion [10]. Moreover, this function has consequences to the entire intestinal epithelial response to injury as supported by previous findings [6,9,39]. Although these findings are highly suggestive, further studies will be necessary to fully determine the exact mechanisms by which DCLK1 in TCs regulate this response.

DCLK1-expressing TC expansion has been observed in human Barrett’s esophagus, chronic gastritis in transgenic mice, rat gastric mucosa and intestinal neoplasia [14,40,41]. While TCs are not usually proliferative, it appears that mutations acquired by stem cells or progenitors can be passed on to TCs, which might then interconvert into tumor initiating cells under inflammatory or injurious conditions. Alternatively, putative “long-lived” TCs might acquire and maintain mutations, finally initiating tumorigenesis after a secondary insult such as colitis [6]. During the early stages of tumorigenesis, DCLK1+ TC expansion is observed in the gastrointestinal niche where they interact with neurons and promote tumorigenesis by secreting acetylcholine to stimulate enteric nerves. Notably, intestinal epithelial cells can express acetylcholine receptors to activate Wnt signaling and regulate the differentiation of intestinal epithelial cells which may be required for tumorigenesis [42]. Using lineage tracing mouse models, Nakanishi et al. and Westphalen et al. concurrently demonstrated the DCLK1+ TC’s cell-of-origin status in Wnt-driven tumorigenesis. In the Nakanishi study, the Apc^Min/+^ model of intestinal polyposis was crossed with a Dclk1^Cre-ERT^ mouse to generate lineage tracing (Apc^Min/+^;Dclk1^Cre-ERT^;R26^LacZ^) and diptheria-toxin receptor TC-specific deletion (Apc^Min/+^;Dclk1^Cre-ERT^;iDTR) mice. Dclk1+ TC-based lineage tracing specifically traced the entirety of the adenoma in these mice. In comparison, an intestinal stem cell marker Lgr5-based model traced the entirety of the normal epithelium and the polyp. Moreover, deletion of DCLK1+ TCs using the diptheria-toxin receptor model resulted in a complete collapse of polyps within days [43]. The Westphalen study made use of an alternative Dclk1^Cre^ model which was crossed to an Apc^flox/flox^ mouse. In this model, spontaneous tumorigenesis did not occur. However, lineage tracing experiments demonstrated a small, but abnormally long-lived, population of DCLK1+ TCs in the intestinal epithelium. In conditions of colitis induced chemically via DSS, these long-lived TCs gave rise to tumors with a severe adenocarcinoma-like phenotype [6]. Importantly, this study was the first to ascertain the existence of multiple functionally unique populations of TCs. This finding has now been confirmed by single-cell RNA-Sequencing studies which identified a separate immunomodulatory population of TCs [44].

In summary, DCLK1-expressing TCs play an important role in stimulating gastrointestinal epithelial stem cells in the microenvironment and contributing to cancer progression [45]. Moreover, studying the two distinct subpopulations of TCs separately may clarify their dual-role in epithelial restitution and tumorigenesis. Promisingly, specific markers for each TC subtype have already been identified [44]. Finally, limited evidence suggests that DCLK1-expressing intestinal TCs in the gut can promote tumor progression in hepatocellular carcinoma (HCC) through activating alternative macrophages in tumor microenvironment via secreting IL-25 [46]. This distant signaling functionality across the gut-liver axis adds an interesting new dimension to understanding the role of TCs.

## 3. Function of DCLK1+ Acinar and Tuft Cells in Pancreatitis and Pancreatic Cancer

In the pancreas, DCLK1 is a marker of a population of pancreatic cancer-initiating cells, some of which have morphological and molecular features of gastrointestinal TCs [47]. However, DCLK1 also notably marks pancreatic acinar cells, which are a likely source of tumorigenesis through the acinar-ductal metaplasia process. Genetic lineage tracing experiments show that Dclk1+ pancreatic epithelial cells are necessary for pancreatic regeneration following injury and chronic inflammation. Moreover, KRAS mutation in Dclk1+ pancreatic epithelial cells leads to pancreatic cancer in the presence of induced pancreatitis [43]. In pancreatic tumors, it has recently been shown that immune cell-derived IL-17 regulates the development of TCs via increased expression of DCLK1, POU domain class 2 transcription factor 3 (POU2F3), aldehyde dehydrogenase 1 family member A1 (ALDH1A1), and IL17RC [48]. Intriguingly, DCLK1 kinase inhibitor can inhibit DCLK1+ organoids derived from pancreatic ductal adenocarcinoma patient tumors, indicating that DCLK1 activity and perhaps DCLK1+ TCs or acinar cells are a potential target for pancreatic ductal adenocarcinoma [49]. Although the underlying signaling mechanisms of DCLK1+ epithelial cell-mediated tumorigenesis require further elaboration, DCLK1 and DCLK1+ epithelial cells such as TCs are likely to be a target for new classes of immunotherapies and TME-remodeling drugs in gastrointestinal tract cancers.

## 4. Interactions between DCLK1 and the Tumor Microenvironment

CSCs depend on the surrounding microenvironment to maintain immune evasion, EMT, drug efflux, DNA repair, signaling pathway regulation, metabolic reprogramming, and epigenetic reprogramming to enhance tumor metastasis, multi-drug resistance and antitumor immunity [50]. Hypoxia is a key regulator of the TME and evidence indicates that DCLK1-positive colorectal cancer cells have increased stemness in hypoxic conditions [51]. Hypoxia famously induces CSCs to express hypoxia inducible factor (HIF) which is a key factor in inducing vascular endothelial growth factor (VEGF) and angiogenesis, which in turn further fuel CSCs. Knockdown of DCLK1 with siRNA or downregulation of DCLK1 with a kinase inhibitor (XMD8-92) results in decreased expression of angiogenic markers/VEGF receptors (VEGFR1 and VEGFR2) and EMT-related transcription factors ZEB1, ZEB2, Snail and Slug [19,52] in pancreatic tumor xenografts. Hypoxia can also be increased epigenetically by histone lysine demethylase 3A (KDM3A) overexpression in pancreatic cancer cells, which leads to increased expression of DCLK1. Knockdown of KDM3A in this context results in reduced invasion, spheroid formation, and orthotopic tumor formation [53]. Finally, in renal cell carcinoma, siRNA-mediated knockdown of DCLK1 significantly sensitized co-cultured endothelial cells to the vascular endothelial growth factor receptor (VEGFR) inhibitor sunitinib in an in vitro angiogenesis assay, demonstrating that expression of DCLK1 on neoplastic cells directly modulates this component of the tumor microenvironment. However, further studies will be needed to determine if this effect is direct [17]. Together these results link the activity of DCLK1, hypoxia, and angiogenesis and further research should be promising.

EMT and CSCs are both linked by key biological characteristics, such as resistance to cytotoxic T lymphocytes (CTLs) and reliance on TGF-β signaling pathway [54]. Microenvironmental changes, such as hypoxia, induce CSCs and in turn CSCs maintain plasticity in their niche via inflammation, EMT, and hypoxia through various signaling pathways [55]. Strong evidence demonstrates that DCLK1 is a regulator of EMT in gastric, colorectal, pancreatic, breast, renal, and other cancers [56,57]. EMT is defined as cellular phenotypic changes from epithelial to mesenchymal type with high expression of N-cadherin and Vimentin [58], and is further associated with TGF-β signaling pathway and functional migration, invasion, metastasis, extracellular matrix (ECM) alteration, apoptosis and drug resistance [59]. Enhanced EMT features after exposure to inflammatory cytokines (i.e., TGF-β, interferon gamma (IFN-γ) and TNF-α) can impact proliferation, differentiation and apoptosis of natural killer cells (NKs) and T and B cells [60], suggesting the importance of EMT in the tumor immune microenvironment. Indeed, evidence suggests that blocking TGF-β signaling may sensitize tumors to immune checkpoint inhibitors [61], and checkpoint ligand programmed cell death 1 (PD-1) ligand (PD-L1) is frequently upregulated in EMT-high tumors.

Additionally, miRNAs are known to play an important role in regulating EMT [62]. Members of miR-200 family directly inhibit ZEB1/ZEB2 activity and overexpression of the miR-200 family can suppress EMT and sensitize cancer cells to chemotherapeutic agents [63]. Knockdown of DCLK1 expression leads to down-regulation of miR-200a, miR-144, and miR-let7a along with downregulation of EMT-associated transcription factors ZEB1, ZEB2, Snail, Slug, and Twist in human pancreatic and colon cancer cells [20,64]. Therefore, DCLK1-mediated EMT could be a target in decreasing HIF levels to regulate angiogenesis and suppress migration by inhibiting cell-to-cell adhesion. Moreover, targeting DCLK1-mediated EMT to regulate TGF-β pathway may alter resistance to CTLs and other anti-tumor immune cells. Blocking DCLK1-mediated EMT also may damage CSC homeostatic processes through several correlated signaling pathways. A study using lung cancer models showed that downregulation of miR-200 family members and upregulation of ZEB1 not only drive EMT, but also lead to upregulation of the PD-L1 in association with exhaustion of intratumoral CD8+ T lymphocytes, which ultimately promotes metastasis [65]. These results suggest that targeting DCLK1-mediated EMT may increase PD-L1 regulated CD8+ T lymphocyte infiltration via regulation of the miR-200 family. Indeed, some evidence shows that DCLK1 marked CSCs support growth, metastasis, and escape from eradication in the tumor microenvironment [25,66,67]. These results are not the only ones demonstrating a relationship between DCLK1 and miRNA activity. Razi et al. showed that DCLK1 is expressed at higher levels in colorectal cancer (CRC) tissue compared to pre-cancerous polyps and that it is inversely correlated with the expression of functional tumor suppressor miRNAs miR-137 and miR-15a. The combined effect of miR-137/miR-15a loss could be significant in CRC as loss of the first is associated with more severe pathological characteristics, and loss of the second has anti-apoptotic, pro-proliferative, and pro-invasive effects [68].

Another key area of focus for DCLK1′s role in the tumor microenvironment involves its basic activity in cell signal transduction. Unlike other prominent target kinases, little is known about DCLK1′s ligands, interacting proteins, and substrates. This perhaps results from the difficulty in studying DCLK1′s complex isoforms, two of which are initiated from an upstream CpG-island regulated promoter (alpha-promoter) and another two of which are initiated from a downstream TATA-box promoter (beta-promoter). However, strides in understanding DCLK1′s basic molecular function have been made in recent years. Notably, DCLK1 has been identified as a potential RAS effector and activator in multiple studies, and DCLK1 expression in pancreatic cancer patients is correlated with RAS downstream signaling pathways ERK, PI3K, and MTOR [49,69,70,71]. DCLK1-AL (transcribed from the α-promoter and characterized by a lengthened C-terminus) can complex with RAS and increase GTP-bound active RAS [71].

Kato et al. showed that loss of the G9a (EHMT2) histone methyl transferase results in a decrease in the number of Dclk1-positive cells and correlated reduction in Erk phosphorylation in mouse pancreatic intraepithelial neoplasia (mPanIN) lesions of a pancreatic cancer mouse model [70], which concurs with findings in the Dclk1^Cre^;Kras^LSL-G12D^ model of pancreatic tumorigenesis [69]. Ferguson et al. also provided evidence for the importance of the interaction between DCLK1 and ERK in a subset of KRAS-mutant pancreatic cancers [49]. In regards to substrates of DCLK1, Liu et al. used the novel and specific inhibitor DCLK1-IN-1 as a tool to identify several candidates including ERK2, GSK3B, CDK1, CDK2, CHK1, and PKACA. Additional potential substrates in nucleic acid processing such as CDK11, MATR3, and DNA topoisomerase 2-beta (TOP2B) were also identified and phosphopeptides including TOP2B, CDK11B, and MATR3 were significantly decreased after treatment with DCLK1-IN-1. Pathway analysis suggested substrate involvement in RNA processing, insulin signaling, ErbB signaling, proteoglycan synthesis, and maintenance of focal adhesion and tight junction pathways [72]. Finally, Koizumi et al. experimentally identified MAP7D1 (microtubule-associated protein 7 domain containing 1) as a substrate of DCLK1 in cortical neurons and the phosphomimetic MAP7D1 fully rescued the impaired callosal axon elongation in neurons after DCLK1 knockdown [73]. All together, these findings are some of the first to unravel DCLK1′s complex molecular mechanisms and may have implications for future translational research and biomarker development.

## 5. Regulation of Immune Checkpoint and Macrophage Polarization by DCLK1

The tumor immune microenvironment (TIME) refers to the microenvironment as it relates to immune cells including: infiltrated-excluded TIME in which there is a relative lack of cytotoxic T lymphocytes in the core location of the tumor; infiltrated–inflamed TIME in which infiltration occurs to a large degree with expression of immune negative regulatory receptor PD-1 of CTLs and inhibitory PD-L1 of leukocytes; and tertiary lymphoid structure TIME, which contains a large number of lymphocytes, including initial and activated T-cells, regulatory T-cells, B cells, and dendritic cells [74]. Overexpression of PD-L1 on tumor cells inhibits the activation of immune cells by binding PD-1 on the surface of T cells after a T-cell receptor binds to cancer cells to promote PD-1 expression. PD-1 and PD-L1 antibodies can block PD-1/PD-L1 co-inhibition signaling and relieve the inhibition of T cells and induce their cytotoxicity. A recent study reported that DCLK1 regulates the level of PD-L1 expression by affecting the corresponding expression level of yes-associated protein (YAP) in the Hippo pathway in pancreatic tumors [26]. These findings concur with others in renal cancer which also show a direct relationship between DCLK1 and PD-L1 expression [26]. CSCs maintain TME stemness as well as increase angiogenesis, which is associated with reduced recognition of T cells and evasion of the immune system via lack of T-cell recognition [75]. PD-L1 perhaps has a key role in helping DCLK1-positive CSCs to evade the immune system, leading to an immune suppressive microenvironment. Moreover, this process may be linked to DCLK1 regulatory activity on EMT, as the EMT process is also strongly associated with immune checkpoint [76].

In the TME, TAMs can serve an anti-tumorigenic or pro-tumorigenic role depending on their status. Typically, the M1 pro-inflammatory macrophage status is correlated to tumor suppression, while the M2 anti-inflammatory/tissue-repair status promotes tumor progression and metastasis. CSCs can secrete various cytokines and chemokines to recruit TAMs to infiltrate the tumor and maintain the M2 phenotype. In turn, M2 macrophages activate STAT3 signaling via secreting IL-6 and epidermal growth factor to increase the expression of Sox2 and enhance the tumorigenic potential of CSCs. In addition, M2 macrophages can also secrete TGF-β to induce EMT and maintain stemness. Overexpression of DCLK1 has been related to worse clinical prognosis via increasing immune and stromal components in colon and gastric cancer patients, and DCLK1 affects multiple immune cell types such as TAMs and Treg and notably inhibits CD8+ T-cells by increasing inhibitor proteins TGF-β1 and chemokine (C-X-C motif) ligand 12 (CXCL12) and their receptors [77]. A recent study demonstrated that overexpression of DCLK1-AL in pancreatic tumor cells can lead to polarization of M1-macrophages towards an M2-phenotype characterized by secretion of chemokines and cytokines such as IL-6, IL-10, and CXCL12, which enhance tumor cell migration, invasion, and self-renewal [28]. In addition, DCLK1-AL induced M2-macrophages inhibited CD8+ T-cell proliferation and Granzyme-B activation, resulting in immunosuppression. Interestingly, silencing DCLK1 caused macrophages to retain the M1 phenotype and abrogated the M2-macrophage ability to enhance aggressiveness and self-renewal in pancreatic cancer cells. Together, these findings suggest that DCLK1 is a promising target to enhance antitumor effect through regulating TIME in some types of cancer (Figure 1).

## 6. Development of DCLK1-Targeted Therapeutic Agents and Biologics

The recent discovery of new CSC surface markers and functional membrane proteins has led to suitable candidate targets such as CD13 and α3β1 for hepatocellular carcinoma (HCC) and bladder cancer, respectively [78,79]. DCLK1 is an optimal target as it represents a more specific CSC marker for colorectal, pancreatic, and possibly other cancers such as gastric cancer, esophageal cancer, breast cancer and renal carcinoma. A small molecule kinase inhibitor, LRRK2-IN-1, was first reported to regulate DCLK1-mediated stemness and EMT by suppressing DCLK1 kinase activity in colorectal and pancreatic cancer [80]. LRRK2-IN-1 impaired cell proliferation, induced apoptosis, and decreased colony formation capacity in cholangiocarcinoma primary cells. Interestingly, it was also shown that DCLK1 marks a subpopulation of LGR5+ and CD133+ CSC-like cells in cholangiocarcinoma, suggesting its potential as a target in this disease [81]. However, LRRK2-IN-1 has notable activity against ERK5 and sub-optimal properties for in vivo delivery. Recently a more specific and in vivo-compatible inhibitor, DCLK1-IN-1, was developed to target the DCLK1 kinase domain based on chemo-proteomic profiling and structure-activity based design. Importantly, this inhibitor showed significant activity against clinically relevant DCLK1+ patient-derived pancreatic ductal adenocarcinoma organoids [49]. Additionally, DCLK1-IN-1 was shown to be effective in CRC using kinase-modified engineered DCLK1 in the DLD-1 cell line [72]. Further studies are needed using DCLK1-IN-1 and other specific DCLK1 kinase inhibitors, and an assessment of its ability to influence anti-tumor immunity is especially desirable as the clinical use of kinase inhibitors in conjunction with immune checkpoint therapies is emerging [82].

Overexpression of DCLK1-AL induces the expression of aldehyde dehydrogenase, stimulates CSC self-renewal, and enhances resistance to FDA-approved receptor tyrosine kinase inhibitors (sunitinib/sorafenib) and mammalian target of rapamyoin inhibitors (everolimus/temsirolimus) in renal cell carcinoma (RCC), suggesting its value as a target in this cancer. A novel monoclonal antibody (CBT-15) was developed to target DCLK1′s extracellular C-terminus and effectively blocked RCC tumorigenesis in an RCC xenograft model [25]. Notably, DCLK1 variants containing the extracellular domain show restricted expression in normal tissue but overexpression in tumor tissue. CBT-15 also showed a significant effect in inhibiting tumor growth in vivo in mouse models of pancreatic cancer [71]. Another DCLK1-targeted mAb, DCLK1-87, stains tissue regions with CSC-marker ALDH expression in CRC, and CRC patients bearing tumors with low tissue staining intensity from this mAb showed improved survival [83].

Utilizing well-characterized CSC markers makes it possible to develop CAR-T cells with the potential to eliminate CSCs. As a CAR-T target, DCLK1 single-chain antibody variable fragment (CBT-511), showed a prominent cytotoxic effect against tumor cells and reduced tumor growth in CRC [27]. CBT-511 also increased IFN-γ release in CRC cells (2D and 3D). It has been reported previously that CSCs decrease the number of activated dendritic cells which is accompanied by decreased secretion of IL-10, IL-12 and IFN-γ cytokines, resulting in the inhibition of proliferation and differentiation of immature T lymphocytes [84]. Further investigation of DCLK1-targeted CAR-T in this context is warranted (Table 1).

## 7. Future Directions for DCLK1 Research and Drug Development

Although traditional radiotherapy and chemotherapy have therapeutic effects on tumors, clinical data show that CSCs are resistant to chemotherapy and radiotherapy, which is a key reason for tumor metastasis and recurrence. Therefore, it is desirable to develop specific and effective targeted therapies against CSCs. DCLK1 is a promising therapeutic target as shown by studies using kinase inhibitor, mAb or CAR-T. However, further studies of DCLK1-expressing TCs and their biological effect in normal conditions and in initiating cancer will be needed to safely target DCLK1. Furthermore, exploration of DCLK1′s relationship to other biological aspects are sorely needed.

Under hypoxia, DCLK1 overexpression induces stemness, but the intermediary mechanisms remain unknown. Hypoxia is known to protect CSCs from chemotherapy and radiation therapy-mediated damage in the TME, and to induce angiogenesis by secreting VEGF and recruiting monocytes, macrophages, macrophages and endothelial cells. Moreover, it limits the proliferation and activation of cytotoxic CD8+ T-cells, activates WNT and Notch signaling pathways to maintain self-renewal, and induces TGF-β signaling to promote EMT. Although DCLK1 is upregulated in angiogenesis and regulates chemotherapy resistance and cancer stemness by WNT signaling, knowledge of the effect of DCLK1 on hypoxia-driven immune cells, endothelial cells, blood vessels, and ECM remains limited [91]. Prior studies show that hypoxia induce CSCs to different metabolic phenotypes including glycolysis for the quiescent M state and oxidative phosphorylation for the proliferative E-state to enhance chemoresistance and acquire other stem-cell characteristics [92,93,94]. Strong evidence shows that the metabolism of CSCs is context-dependent and reliant on glycolysis or mitochondrial oxidative metabolism [95,96,97,98,99,100]. Currently, conventional therapies such as chemotherapy and radiotherapy have a low effect on CSCs because of increased expression of drug transporters, maintenance of a slow dividing state (quiescence), and efficient DNA repair mechanisms. Metabolic phenotypes are directly related with CSC dividing state and it is thought that targeting CSC metabolism may be an effective way to eliminate chemo-resistance and tumor relapse. One interesting study showed that Doublecortin-like (a splice-variant produced from DCLK1′s alpha promoter) knockdown is associated with reduced mitochondrial activity which significantly decreases tumor growth by regulating cytochrome c oxidase activity and ATP synthesis in neuroblastoma tumor xenografts. Another study showed that glycolysis promotes the expression of DCLK1 and maintains the CSC and EMT phenotypes via low reactive oxygen species levels in chemo-resistant pancreatic cancer cells [101]. Tumor microenvironmental factors including hypoxia, glucose deprivation, low pH, oxygen stress, and others are key in promoting CSC selection of metabolic pathways leading to metastasis or drug-resistance [102]. Metabolic alterations may cause cells to acquire stem-cell-like characteristics, and DCLK1+ TCs are long-lived and quiescent before they are activated by injury. Recent studies demonstrate that targeting oxidative phosphorylation may inhibit CSC metabolic processes and proliferation in some cancers [103]. Switching metabolic phenotypes of CSCs and TCs by enhancing oxidative phosphorylation to inhibit their tumorigenesis or tumor growth may be a feasible direction for further study. Future studies should focus on whether targeting DCLK1 to regulate metabolic processes or targeting metabolic activity of DCLK1+ TCs or CSCs may be a viable focus for therapy.

Targeting the TIME has already resulted in remarkable achievements including CAR-T and CAR-NK technologies that can potentially kill CSCs. The composition of immune cells in the tumor microenvironment will affect their response to specific immunotherapies and alter antigen presentation and macrophage polarization. Inhibition of IL-6 secretion from TAMs can inhibit the activation of CSCs to improve therapy, and overexpression of immune checkpoint ligand PD-L1 on CSCs blocks the cytotoxic CD8+ T-cell response [104]. Although existing data is relatively limited, DCLK1 has shown promising prospects in the above areas and a full assessment of DCLK1′s impact on immune checkpoint and pro-tumor macrophages is warranted.

## 8. Conclusions

DCLK1+ TCs are closely related with tumor initiation and chronic inflammatory diseases. Currently, the TC biological effect on cancer initiation and progression is not fully understood. Although CSCs drive chemotherapy and radiotherapy failures, there is still no effective therapeutic strategy against them. Due to these limitations, targeting the tumor microenvironment provides a prospective option for cancer treatment. However, in different tumor types or different developmental stages of the tumor, the interaction between CSCs and the microenvironment varies and will complicate developing these therapeutic strategies. Finding a reliable molecular target is crucial, and DCLK1 is one such potential marker that should be pursued in this context. DCLK1 is a multifaceted target due to several isoforms with variable functions and cellular localization. However, new evidence of their various roles is emerging. Importantly, efforts are underway to determine how DCLK1 functions within the TC to promote the response to injury, including how it modulates the immune microenvironment and how it balances this role with potential pro-tumorigenic signaling. When sufficient knowledge is gained, a variety of DCLK1-specific targeting modalities are already available for translation including specific kinase inhibitors and targeted monoclonal antibodies and CAR-T therapies.

## Figures and Tables

**Figure 1 cancers-12-03801-f001:**
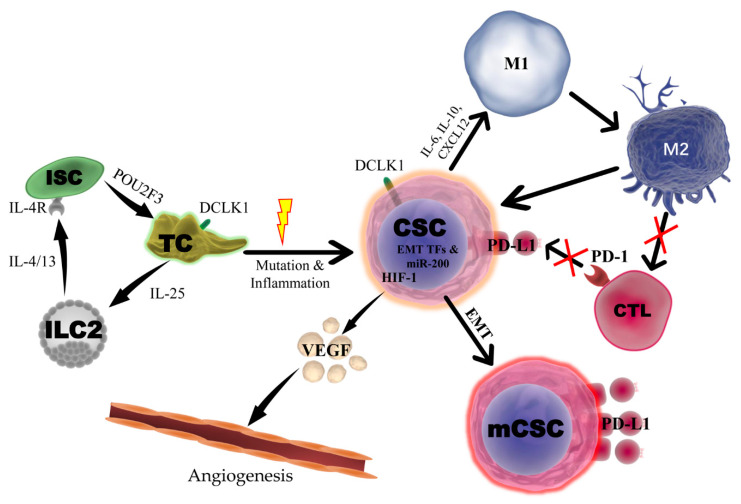
Potential Role of DCLK1 and DCLK1+ Cells in the Intestinal Tumor Microenvironment. DCLK1+ tuft cells (TC) will recruit innate lymphoid type-2 cells (ILC2) by secreting IL-25 which in turn reprograms intestinal stem cells (ISC) through IL4/13-IL4R signaling to express transcription factor POU2F3 leading to TC hyperplasia. In the presence of mutation and inflammation, DCLK1+ TCs can be converted to cancer stem cells (CSC) and initiate a tumor. Under hypoxia, DCLK1+ CSCs may induce angiogenesis by upregulation of HIF-1 and secretion of VEGF. Furthermore, DCLK1+ CSCs promote EMT via the miR-200 family leading to metastatic CSCs (mCSC) with high levels of PD-L1. DCLK1+ CSCs further regulate the immune tumor microenvironment by polarization of M1 macrophages towards an M2 status by secreting IL-6, IL-10 and CXCL12, which leads to inhibition of T-cell proliferation and activation. DCLK1-positive CSCs also express programmed death ligand 1 (PD-L1) expression to inhibit CD8/PD1++ CTL function.

**Table 1 cancers-12-03801-t001:** Investigational targeted therapies against doublecortin-like kinase 1.

Name of Drug	Class of Drug	DCLK1 Affinity	Other Significant Targets	Cancer Types Tested	Level of Evidence	Functional Target of Drug	Author (Year)	PMID
LRRK2-IN-1	Kinase inhibitor	<60 nM	LRRK2, ERK5	CRC, PDAC, CCA	In vitro, in vivo, and ex vivo	Stemness, proliferation, migration, invasion, apoptosis, cell cycle, DNA damage, EMT and tumor growth	Weygant et al. (2014) [80]	24885928
							Kawamura et al. (2017) [85]	29048622
							Nevi et al. (2020) [81]	32978808
							Suehiro et al. (2018) [86]	30396941
XMD8-92	Kinase inhibitor	< 100 nM	ERK5, DCLK2	Mesothelioma, PDAC	In vitro and in vivo	Stemness, EMT, angiogenesis, proliferation and tumor growth	Sureban et al. (2014) [52]	24880079
							Wang et al. (2017) [87]	28560410
DCLK1-IN-1	Kinase inhibitor	< 60 nM	DCLK2	PDAC, CRC	In vitro and ex vivo	Proliferation, invasion and stemness	Ferguson et al. (2020) [49]	32251410
							Ferguson et al. (2020) [88]	32530623
NP-siDCAMKL-1	Nanoparticle-encapsulated siRNA	N/A	None	CRC, HCC, PDAC	In vitro and in vivo	Tumor growth	Sureban et al. (2011) [64]	21929751
							Sureban et al. (2015) [89]	26468984
							Sureban et al. (2013) [19]	24040120
CBT-15	Monoclonal antibody	<1 nM	None	PDAC, RCC	In vitro and in vivo	ADCC and tumor growth	Ge et al. (2018) [25] Qu et al. (2019) [71]	29577277 31467540
DCLK1–HA–PEG–PLGA	Bifunctional-antibody/nanoparticle conjugate	N/A	CD44	Breast cancer	In vitro and in vivo		Qiao et al. (2016) [90]	27994463
CBT-511	Chimeric antigen receptor T-cells	<1 nM	None	CRC	In vitro and in vitro	Proliferation and tumor growth	Sureban et al. (2019) [27]	31878090

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
