# Peer review of "Tuft and Cancer Stem Cell Marker DCLK1: A New Target to Enhance Anti-Tumor Immunity in the Tumor Microenvironment"

_cancers, 2020, doi:10.3390/cancers12123801_

Round 1

Reviewer 1 Report

In the manuscript entitled “Tuft and cancer stem cell marker DCLK1: a new target to enhance anti-tumor immunity in the tumor microenvironment” authors   reviewed the mechanistic role of DCLK1 in the tumor microenvironment, discussing the potential for targeting DCLK1 in colon, pancreatic and renal cancer.

The review is well structured and rich in content, although there are still very few studies in the literature.

We only suggest the inclusion and discussion of two very recent articles:

1: Nevi L, Di Matteo S, Carpino G, Zizzari I, Safarikia S, Ambrosino V, Costantini D, Overi D, Giancotti A, Monti M, Bosco D, De Peppo V, Oddi A, De Rose AM, Melandro F, Bragazzi MC, Faccioli J, Massironi S, Grazi GL, Panici PB, Berloco PB, Giuliante F, Cardinale V, Invernizzi P, Caretti G, Gaudio E, Alvaro D. DCLK1, a putative novel stem cell marker in human cholangiocarcinoma. Hepatology. 2020 Sep 25.  

2: Razi S, Sadeghi A, Asadi-Lari Z, Tam KJ, Kalantari E, Madjd Z. DCLK1, a promising colorectal cancer stem cell marker, regulates tumor progression and invasion through miR-137 and miR-15a dependent manner. Clin Exp Med. 2020 Sep 23.  

Author Response

We are highly appreciative to the Reviewer for taking the time to carefully assess our manuscript and provide these important suggestions. We have updated the text to discuss these new findings concerning DCLK1’s role as a CSC marker in cholangiocarcinoma and its interactions with miR-137 and miR-15a. The changes to the text are as listed below:

Lines 235-241 now state in reference to DCLK1/miR-137/15a interactions:

“These results are not the only ones demonstrating a relationship between DCLK1 and miRNA activity. Razi et al showed that DCLK1 is expressed at higher levels in CRC tissue compared to pre-cancerous polyps and that it is inversely correlated with the expression of functional tumor suppressor miRNAs miR-137 and miR-15a. The combined effect of miR-137/miR-15a loss could be significant in CRC as loss of the first is associated with more severe pathological characteristics, and loss of the second has anti-apoptic, pro-proliferative, and pro-invasive effects [68].”

Lines 332-335 now state in reference to DCLK1’s status as a marker in cholangiocarcinoma:

“LRRK2-IN-1 impaired cell proliferation, induced apoptosis, and decreased colony formation capacity in cholangiocarcinoma primary cells. Interestingly, it was also shown that DCLK1 marks a subpopulation of LGR5+ and CD133+ CSC-like cells in cholangiocarcinoma, suggesting its potential as a target in this disease [82].”

Reviewer 2 Report

Extensive review on DCLK1 marker, its role in cancer and as a potential target.  The review is well written, clear, well documented.

Author Response

We appreciate the Reviewer for spending their time in assessing the quality of this manuscript and are grateful for the positive assessment regarding this important topic.

Reviewer 3 Report

The present paper describes the importance of the expression of the DCKL1 marker on TUFT cells and on Cancer stem cells in relation to the process of tumorigenesis further suggesting that DCKL1might represent a potential therapeutic target in several tumors types.

Although the paper is interesting and up to date I find some passages too long and confusingly written.

I therefore suggest a critical revision of the whole manuscript trying to better define the most important concepts relating to i) DCLK1+ tuft cells and CSC, ii) DCLK1 and TME, ii) DCLK1 role in macrophage polarization, iii) DCLK1 as a target for therapy

Moreover since TUFT cells can play a double role within the intestinal epithelium as they can stimulate the repair after injury or, instead, they can promote tumor formation, I suggest to better emphasize and explain evidences showing that these cells can perform this dual function and the DCLK1 involvement.

Please rewrite:

- lines 86-90. ….immunotherapy (CAR-T) therapies………novel cancer immunotherapies…..?

- line 165 …..that tumor DCLK1 expression modulates this component of the tumor ….. :

Perhaps “ that the expression of DCLK1 on neoplastic cells  directly modulates  (or inhibits??) the release of this growth factor within the tumor microenviroment”.

-lines 253-254 : I understand the meaning but I think it is unclear.

I suggest to write: Recently the discovery of specific CSC markers or functional membrane proteins have suggested that CD13 and a3b1 are suitable candidate targets in HCC and in bladder cancer respectively.

So many points need to be explained more clearly .

Author Response

We thank the Reviewer for their time in assessing this manuscript and bringing our attention to these important points. We have endeavored to adjust and correct the manuscript accordingly and hope the result is satisfactory. Specifically, we have gone through the concepts the Reviewer listed as lacking in the manuscript and critically revised to improve clarity where possible. Moreover we have also significantly expanded discussion of the dual role of tuft cells and cited relevant literature. Although tuft cells maintain a dual role in the intestinal epithelium whereby they may both stimulate its repair after sensing damage and signaling ILC2s or may serve as a cell-of-origin for tumor initiation, the role of DCLK1 in this process remains unclear. This is because studies to assess this functionality have been focused on drastic phenotypes such as deleting tuft cells using engineered diphtheria toxin receptor mice or preventing their lineage commitment via POU2F3 transcription factor knockout. However, there is evidence that DCLK1 plays a role in these processes. Knockout of DCLK1 alpha-promoter isoforms (which are exclusively expressed in tuft cells) in the intestinal epithelium of mice leads to severe barrier deficits in the presence of inflammatory colitis induced by DSS. Indeed the effect is so severe that experimental mice may die as a result of this deficit. Although this doesn’t directly implicate DCLK1 in this function, it is highly suggestive. One recent and important study (Yi et al 2019) further supports this idea. In this study Yi and colleagues knocked out DCLK1 alpha-promoter isoforms in the mucin-type O-glycan deficient mouse model of ulcerative colitis (UC). DCLK1-KO UC mice developed more severe colitis due to a deficit in epithelial proliferative capacity which was phenotypically characterized by pathological symptoms including an increase in immune infiltration. Notably, DCLK1-KO UC mice showed an expansion of tuft cells just like wild-type controls. These findings demonstrate that tuft cells depend on DCLK1 expression, and not just TC expansion, to mount an effective response to infection and initiate proper epithelial restitution. Although further studies are needed to fully determine the mechanisms by which DCLK1 in tuft cells regulate this response, we believe it is important to maintain awareness of this aspect of DCLK1/tuft cell biology especially when discussing ways to target it in cancer for its CSC/pro-tumorigenic role. Improper targeting modalities are likely to cause side-effects through affecting DCLK1’s role in this process, and future DCLK1-targeted therapies may be contraindicated in patients with a history of UC or other related gastrointestinal conditions. The relevant changes to the text are as follows (lines 117-130):

“There is strong evidence that DCLK1 in TCs play an important functional role in epithelial repair processes of the gut. Intestinal epithelium-specific knockout of DCLK1 (VilCre;Dclk1flox/flox) leads to increased severity of injury and death in mouse whole body irradiation and dextran sulfate sodium (DSS)-induced colitis models [6,9,39]. A recent study expounded on this idea more directly. Yi et al reported the deletion of DCLK1 in the mucin-type O-glycan deficient model of ulcerative colitis (UC) resulted in greater severity of disease characterized by enhanced mucosal thickening and increased inflammatory cell infiltration. They found that in the absence of DCLK1, epithelial proliferative responses to chronic inflammation were impaired. However, the deletion of DCLK1 did not affect the numbers of intact TCs. These results indicate that DCLK1 expression is a regulator of TC activation status, despite not being involved in TC expansion [10]. Moreover, this function has consequences to the entire intestinal epithelial response to injury as supported by previous findings [6,9,39]. Although these findings are highly suggestive, further studies will be necessary to fully determine the exact mechanisms by which DCLK1 in TCs regulate this response.

As for the evidence of the tuft cell role as a potential CSC, there are a variety of studies demonstrating this in mouse models and supportive evidence in human cancer. The first study to demonstrate the tuft cells CSC capacity was reported in Nature Genetics and used the ApcMin/+ model of adenoma crossed to a Dclk1Cre-ERT2 mouse. The lineage tracing capabilities of this mouse allowed confirmation that Dclk1+ cells specifically were the cell-of-origin for adenomas. Furthermore, deletion of the tuft cell using engineered diphtheria toxin receptor (ApcMin/+;Dclk1Cre-ERT2;R26DTR) resulted in a complete and total collapse of polyps within days. This study was soon followed up with confirmatory evidence in a Dclk1Bac-Cre;Apcfl/fl model. Unlike the ApcMin/+ based model, this model did not support the idea that Apc-loss alone in the tuft cell was sufficient to induce tumorigenesis. Indeed, the addition of inflammation via dextran sulfate sodium (DSS) was necessary to cause this effect. Additionally, this study introduced the concept of two populations of DCLK1+ tuft cells – a proliferative population and a quiescent/long-lived population that was the likely source of tumorigenesis. It is interesting to note that the existence of multiple populations of tuft cells has now been confirmed using single-cell RNA-Sequencing. It has been proposed that the less common population, which may be consistent with the long-lived population, has an immunomodulatory function. If this proves true, this would clarify a double-edged potential for tuft cells in the inflammatory epithelial microenvironment. The changes to the text are as follows (lines 131-165):

“DCLK1-expressing TC expansion has been observed in human Barrett’s esophagus, chronic gastritis in transgenic mice, rat gastric mucosa and intestinal neoplasia mice [14,40,41]. While TCs are not proliferative, it appears that mutations acquired by stem cells or progenitors can be passed on to TCs, which might then interconvert into tumor initiating cells under inflammatory or injurious conditions. Alternatively, putative “long-lived” TCs might acquire and maintain mutations, finally initiating tumorigenesis after a secondary insult such as colitis [6]. During early stages of tumorigenesis, DCLK1+ TCs expansion is observed in the gastrointestinal niche where they interact with neurons and promote tumorigenesis by secreting acetylcholine to stimulate enteric nerves. Notably, intestinal epithelial cells can express acetylcholine receptors to activate Wnt signaling and regulate the differentiation of intestinal epithelial cells which may be required for tumorigenesis [42]. Using lineage tracing mouse models, Nakanishi et al and Westphalen et al concurrently demonstrated the DCLK1+ TC’s cell-of-origin status in Wnt-driven tumorigenesis. In the Nakanishi study, the ApcMin/+ model of intestinal polyposis was crossed with a Dclk1Cre-ERT mouse to generate lineage tracing (ApcMin/+;Dclk1Cre-ERT;R26LacZ) and diptheria-toxin receptor TC-specific deletion (ApcMin/+;Dclk1Cre-ERT;iDTR) mice. Dclk1+ TC-based lineage tracing specifically traced the entirety of the adenoma in these mice. In comparison, an intestinal stem cell marker Lgr5-based model traced the entirety of the normal epithelium and the polyp. Moreover, deletion of DCLK1+ TCs using the diptheria-toxin receptor model resulted in a complete collapse of polyps within days [43]. The Westphalen study made use of an alternative Dclk1Cre model which was crossed to an Apcflox/flox mouse. In this model spontaneous tumorigenesis did not occur. However, lineage tracing experiments demonstrated a small, but abnormally long-lived population of DCLK1+ TCs in the intestinal epithelium. In conditions of colitis induced chemically via DSS, these long-lived TCs gave rise to tumors with a severe adenocarcinoma-like phenotype [6]. Importantly, this study was the first to ascertain the existence of multiple functionally unique populations of TCs. This finding has now been confirmed by single-cell RNA-Sequencing studies which identified a separate immunomodulatory population of TCs [44].

In summary, DCLK1 expressing TCs play an important role in stimulating gastrointestinal epithelial stem cells in the microenvironment and contributing to cancer progression [45]. Moreover, studying the two distinct subpopulations of TCs separately may clarify their dual-role in epithelial restitution and tumorigenesis. Promisingly, specific markers for each TC subtype have already been identified [44]. Finally, limited evidence suggests that DCLK1 expressing intestinal TCs in the gut can promote tumor progression in hepatocellular carcinoma (HCC) through activating alternative macrophages in tumor microenvironment via secreting IL-25 [46]. This distant signaling functionality across the gut-liver axis adds an interesting new dimension to understanding the role of TCs.”

Minor Issues:

Please rewrite:

- lines 86-90. ….immunotherapy (CAR-T) therapies………novel cancer immunotherapies…..?

- line 165 …..that tumor DCLK1 expression modulates this component of the tumor ….. :

Perhaps “ that the expression of DCLK1 on neoplastic cells  directly modulates  (or inhibits??) the release of this growth factor within the tumor microenviroment”.

-lines 253-254 : I understand the meaning but I think it is unclear.

I suggest to write: Recently the discovery of specific CSC markers or functional membrane proteins have suggested that CD13 and a3b1 are suitable candidate targets in HCC and in bladder cancer respectively.

Response: We apologize about confusion related to these passages. We have modified them according the Reviewer’s suggestion and hope their meaning is more clear.

Lines 86-90 (now Lines 89-94) have been changed to state:

“Furthermore, several recent studies show that DCLK1 affects tumor growth and metastasis via regulating TAM and immune checkpoint. Finally, monoclonal antibodies and chimeric antigen receptor T-Cells (CAR-T) based on DCLK1 have demonstrated potential as novel cancer immunotherapies [26-28].”

Line 165 (now Lines 199-203) have been changed to:

“Finally, in renal cell carcinoma, siRNA-mediated knockdown of DCLK1 significantly sensitized co-cultured endothelial cells to VEGFR inhibitor sunitinib in an in vitro angiogenesis assay, demonstrating that expression of DCLK1 on neoplastic cells directly modulates this component of the tumor microenvironment. However, further studies will be needed to determine if this effect is direct [17].”

Lines 253-254 (now Lines 326-327) have been modified to:

“The recent discovery of new CSC surface markers and functional membrane proteins has led to suitable candidate targets such as CD13 and α3β1 for HCC and bladder cancer respectively [79,80].”

Reviewer 4 Report

This paper is well-written, focusing on DCLK1 as a possible target for the treatment of cancers. The biochemical mechanisms of DCLK1 activation and the physiological role of DCLK1 in cell functions should be described in more detail, which may be helpful for the readers to understand well the significant role of DCLK1.

The authors may want to describe the specific ligands that activate DCLK1, the activation mechanisms, the localization of DCLK1, the downstream substrates phosphorylated by DCK1, and the final functional events.

Author Response

We are pleased about the overall positive assessment of the manuscript as well as the insightful suggestions from the Reviewer. We believe that the additions suggested by the Reviewer have improved the value of the review article and will be helpful to readers wanting to learn more about this target. Although current understanding of the DCLK1 ligands, activation, and downstream substrates is limited we believe that we have now comprehensively assessed these issues and linked them to cellular localization and downstream functional events where possible. The relevant changes to the manuscript can be found on lines 242-269 and are as follows:

“Another key area of focus for DCLK1’s role in the tumor microenvironment involves its basic activity in cell signal transduction. Unlike other prominent target kinases, little is known about DCLK1’s ligands, interacting proteins, and substrates. This perhaps results from the difficulty in studying DCLK1’s complex isoforms, 2 of which are initiated from an upstream CpG-island regulated promoter (alpha-promoter) and another 2 of which are initiated from a downstream TATA-box promoter (beta-promoter). However, strides in understanding DCLK1’s basic molecular function have been made in recent years. Notably, DCLK1 has been identified as a potential RAS effector and activator in multiple studies, and DCLK1 expression in pancreatic cancer patients is correlated with RAS downstream signaling pathways ERK, PI3K, and MTOR [47,66-68]. DCLK1-AL (transcribed from the α-promoter and characterized by a lengthened C-terminus) can complex with RAS and increase GTP-bound active RAS [68]. Kato et al showed that loss of the G9a (EHMT2) histone methyl transferase results in a decrease in the number of Dclk1-positive cells and correlated reduction in Erk phosphorylation in mPanIN lesions of a pancreatic cancer mouse model [67], which concurs with findings in the Dclk1Cre;KrasLSL-G12D model of pancreatic tumorigenesis [66]. Ferguson et al also provided evidence for the importance of the interaction between DCLK1 and ERK in a subset of KRAS-mutant pancreatic cancers [47]. In regards to substrates of DCLK1, Liu et al used the novel and specific inhibitor DCLK1-IN-1 as a tool to identify several candidates including ERK2, GSK3B, CDK1, CDK2, CHK1, and PKACA. Additional potential substrates in nucleic acid processing such as CDK11, MATR3, and DNA topoisomerase 2-beta (TOP2B) were also identified and phosphopeptides including TOP2B, CDK11B, and MATR3 were significantly decreased after treatment with DCLK1-IN-1. Pathway analysis suggested substrate involvement in RNA processing, insulin signaling, ErbB signaling, proteoglycan synthesis, and maintenance of focal adhesion and tight junction pathways [69]. Finally, Koizumi et al experimentally identified MAP7D1 (microtubule-associated protein 7 domain containing 1) as a substrate of DCLK1 in cortical neurons and the phosphomimetic MAP7D1 fully rescued the impaired callosal axon elongation in neurons after DCLK1 knockdown [70]. All together, these findings are some of the first to unravel DCLK1’s complex molecular mechanisms and may have implications for future translational research and biomarker development.”

Reviewer 5 Report

I the present work the authors have reviewed the literature for the role of DCLK1 a tuft cell and cancer stell cell marker, as a new target that can enhance anti-tumor immunity in the tumor microenvironment.

Their article is very well organized, well-written and has merit for publication after some minor comments.

Please state the rationale behind their choice of molecule. Why did the authors choose the specific molecule for reviewing its role in tumor microenvironment.

In Table I, please add the authors' first name and year along with the Pubmed ID.

They also should highlight their findings at the end of the manuscript.

Author Response

We appreciate the Reviewer’s comments and suggestions and have addressed each point below.

Please state the rationale behind their choice of molecule. Why did the authors choose the specific molecule for reviewing its role in tumor microenvironment.

DCLK1 was chosen as a focus for this article because of its emerging importance in the pre-cancer and cancer TME. Additionally, its role in both the tuft cell and the CSC remain mysterious and we wanted to summarize the literature regarding this topic for other researchers.

In Table I, please add the authors' first name and year along with the Pubmed ID.

We have amended Table I as suggested.

They also should highlight their findings at the end of the manuscript (lines 420-434).

We have expanded the concluding paragraph to better highlight the key points of this manuscript from both a basic and translational perspective. The new text is as follows:

“DCLK1+ TCs are closely related with tumor initiation and chronic inflammatory diseases. Currently, the TC biological effect on cancer initiation and progression is not fully understood. Although CSCs drive chemotherapy and radiotherapy failures, there is still no effective therapeutic strategy against them. Due to these limitations, targeting the tumor microenvironment provides a prospective option for cancer treatment. However, in different tumor types or different developmental stages of the tumor, the interaction between CSCs and the microenvironment varies and will complicate developing these therapeutic strategies. Finding a reliable molecular target is crucial, and DCLK1 is one such potential marker that should be pursued in this context. DCLK1 is a multifaceted target due to several isoforms with variable functions and cellular localization. However, new evidence of their various roles are emerging. Importantly, efforts are underway to determine how DCLK1 functions within the TC to promote the response to injury, including how it modulates the immune microenvironment and how it balances this role with potential pro-tumorigenic signaling. When sufficient knowledge is gained, a variety of DCLK1-specific targeting modalities are already available for translation including specific kinase inhibitors and targeted monoclonal antibodies and CAR-T therapies.”

Round 2

Reviewer 4 Report

This paper is appropriately revised.